# Thrombocytosis and Effects of IL-6 Knock-Out in a Colitis-Associated Cancer Model

**DOI:** 10.3390/ijms21176218

**Published:** 2020-08-27

**Authors:** Valeria Josa, Szilamer Ferenczi, Rita Szalai, Eniko Fuder, Daniel Kuti, Krisztina Horvath, Nikolett Hegedus, Tibor Kovacs, Gergo Bagamery, Balazs Juhasz, Zsuzsanna Winkler, Daniel S. Veres, Zsombor Zrubka, Domokos Mathe, Zsolt Baranyai

**Affiliations:** 1Jahn Ferenc Del-pesti Korhaz es Rendelointezet, Department of Otorhinolaryngology and Head and Neck Surgery, 1135 Budapest, Hungary; 2Laboratory of Molecular Neuroendocrinology, Institute of Experimental Medicine, 1083 Budapest, Hungary; ferenczi.szilamer@koki.mta.hu (S.F.); kuti.daniel@koki.mta.hu (D.K.); horvath.krisztina@koki.mta.hu (K.H.); juhasz.balazs@koki.mta.hu (B.J.); winkler.zsuzsanna@koki.mta.hu (Z.W.); 3Faculty of Medicine, Semmelweis University, 1085 Budapest, Hungary; szrituska@gmail.com; 4Department of Pathology, Uzsoki utcai Hospital, 1145 Budapest, Hungary; fudereniko@gmail.com; 5Department of Biophysics and Radiation Biology, Semmelweis University, 1094 Budapest, Hungary; hegedus.nikolett@med.semmelweis-univ.hu (N.H.); veres.daniel@med.semmelweis-univ.hu (D.S.V.); mathe.domokos@med.semmelweis-univ.hu (D.M.); 6CROmed Translational Research Ltd., 1094 Budapest, Hungary; 7Department of Biophysics and Radiation Biology, University of Pannonia, Institute of Radiochemistry and Radioecology, 8200 Veszprém, Hungary; kt@almos.uni-pannon.hu; 8Mediso Medical Imaging Systems Ltd., 1037 Budapest, Hungary; gergo.bagamery@mediso.hu; 9University Research, Innovation and Service Center, University of Óbuda, 1034 Budapest, Hungary; zsombor.zrubka@uni-corvinus.hu; 10Hungarian Center for Excellence in Molecular Medicine, 6723 Szeged, Hungary; 111st Department of Surgery, Semmelweis University, 1082 Budapest, Hungary; barazso@gmail.com

**Keywords:** thrombocytosis, colorectal cancer, interleukin-6, tumor model, colitis-associated cancer

## Abstract

There is an increasing number of studies showing that thrombocytosis—accompanying a variety of solid tumors including colorectal cancer (CRC)—is associated with shorter survival and earlier development of metastases. The mechanisms of cancer-associated thrombocytosis are not completely understood yet. The aim of our study was to evaluate the role of IL-6 in tumor development and thrombocytosis in mice with inflammation-induced CRC, using a CRISPR/cas9 IL-6 knockout (KO) strain. Adult male FB/Ant mice (*n* = 39) were divided into four groups: (1) IL-6 KO controls (*n* = 5); (2) IL-6 KO CRC model group (*n* = 18); (3) Wild-type (WT) controls (*n* = 6); and (4) WT CRC model group (*n* = 10). CRC model animals in (2) and (4) received azoxymethane (AOM)/dextran sodium sulfate (DSS) treatment to induce inflammation-related CRC. Plasma and liver tissues were obtained to determine platelet counts, IL-6 and thrombopoietin-1 (TPO) levels. In 1 WT and 2 IL-6 KO mice in vivo confocal endomicroscopy and 18F-fluorodeoxyglucose (FDG) PET/MRI examinations were performed to evaluate the inflammatory burden and neoplastic transformation. At the end of the study, tumorous foci could be observed macroscopically in both CRC model groups. Platelet counts were significantly elevated in the WT CRC group compared to the IL-6 KO CRC group. TPO levels moved parallelly with platelet counts. In vivo fluorescent microscopy showed signs of disordered and multi-nuclear crypt morphology with increased mucus production in a WT animal, while regular mucosal structure was prominent in the IL-6 KO animals. The WT animal presented more intense and larger colonic FDG uptake than IL-6 KO animals. Our study confirmed thrombocytosis accompanying inflammation-related CRC and the crucial role of IL-6 in this process. Significantly higher platelet counts were found in the WT CRC group compared to both the control group and the IL-6 KO group. Concomitantly, the tumor burden of WT mice was also greater than that of IL-6 KO mice. Our findings are in line with earlier paraneoplastic IL-6 effect suggestions.

## 1. Introduction

Colorectal cancer (CRC) is among the most prevalent malignancies worldwide. With 1.65 million new cases and nearly 835 thousand deaths in the industrial world in 2015, CRC is the third most common form of neoplasia among males and the second most frequent one among females [1]. When discovered at an early stage, CRC is often curable. The mortality rate from CRC has been following a declining trend since the mid-eighties. This is attributable to screening, which often reveals large bowel lesions before they transform into a tumor and to advances in therapy. As shown by an increasing number of studies, thrombocytosis accompanying a variety of solid tumors including CRC is associated with shorter survival and an earlier development of metastases [2,3,4]. Thrombocytosis in early CRC has even been proposed as a predictive biomarker by one study [4]. However, the mechanisms between cancer-associated thrombocytosis and aggravating systemic CRC are not completely understood. According to some hypotheses, platelets contribute to the formation of metastases by cloaking circulating tumor cells and thereby protecting the latter from mechanical injuries [5,6], as well as from the immune defenses of the body [7,8]. At the same time platelets become activated while traveling through the blood vessels of the tumor. The activated platelets facilitate tumor cell proliferation by secreting a number of angiogenic tumor growth factors, such as thrombopoietin (TPO), platelet factor 4 (PF4), transforming growth factor beta (TGFb), vascular endothelial growth factor (VEGF) and platelet-derived growth factor (PDGF) [9,10,11]. Angiogenesis accelerates the growth of the tumor [12,13,14]. Activated platelets release microvesicles that enhance the invasive potential of cancer cells [7]. Recently, a paraneoplastic signaling pathway has attracted much attention, albeit in ovarian cancer. According to the hypothesis by Stone et al., tumors increase interleukin-6 (IL-6) levels which augments TPO production in the liver. This in turn stimulates megakaryocytes in the bone marrow and eventually leads to thrombocytosis [15]. The increase in TPO through IL-6, coupled with thrombocytosis has not yet been confirmed in CRC. We thus investigated the role of IL-6 in IL-6 gene knockout (KO) and in wild-type (WT) mice with inflammation-induced CRC. We decided to apply a widely used model of colitis-associated cancer (CAC) [16,17] whereby the cancer- inducing inflammatory stage is followed by progression of CRC. In this model, several interleukins including more prominently, IL-6 have been observed to drive progression and invasion [18,19].

## 2. Results

### 2.1. Verification of IL-6 Gene Knockout

We used the standard lipopolysaccharide (LPS)-provocation test which is routinely performed for ascertaining the loss of function of the IL-6 gene. Thirty-nine mice altogether were studied in the following four groups: (1) IL-6 KO controls (*n* = 5); (2) IL-6 KO treatment group (*n* = 18); (3) WT controls (*n* = 6); and (4) the WT treatment group (*n* = 10). In the IL-6 KO group, IL-6 concentration did not increase after LPS administration, whereas WT animals showed a significant elevation of plasma IL-6 (Appendix A).

### 2.2. Histopathological and Clinical Characteristics

After termination, multifocal tumors exhibiting polypoid growth were detected in azoxymethane (AOM)/dextran sodium sulfate (DSS)-treated CRC model mice, mainly in the descendent colon. The number of foci ranged from 2 to 8, with equal of variances (F8,8 = 1.11, *p* = 0.88) and normal distribution in the IL-6 KO and WT subgroups (Shapiro–Wilk test *p* = 0.88 and *p* = 0.88, respectively). Mean (±SD) number of tumor foci were significantly greater (Student’s t-test, t16 = −2.79, *p* = 0.013) in the WT (6.22 ± 0.52) than in the IL-6 KO subgroup (4.22 ± 0.49) (Figure 1A). Mean (±SD) total tumor volume (TTV) was 13.6 (±6.2) mm^3^ (median 16.2, IQR: 9.3–17.3) in the IL-6 KO, while it was 21.9 (±6.6) mm^3^ (median 18.1, IQR: 16.2–28.3) in the WT subgroup. The distribution of TTV was normal in the IL-6-KO subgroup (Shapiro–Wilk test *p* = 0.475), but not in the WT subgroup (Shapiro–Wilk test *p* = 0.014). Despite the greater mean TTV in the WT animals, the applied nonparametric tests did not demonstrate significant differences between the two subgroups (Kolmogorov–Smirnov test exact *p =* 0.352; Mann–Whitney *p* = 0.058, equality of medians test exact *p* = 0.157) (Figure 1B).

The 1st, 2nd, 3rd and 4th TTV quartiles comprised of 4 (TTV range 5.1–10.9 mm^3^), 5 (TTV range 14.6–16.4 mm^3^), 4 (TTV range 17.0–19.6 mm^3^) and 5 (TTV range 22.4–29.7 mm^3^) animals, respectively. Likewise, despite the shift of distribution towards greater TTV quartiles (e.g., larger tumors with more foci) in the WT subgroup (0.0%, 33.3%, 22.2%, 44.4% of animals in the 1st, 2nd, 3rd and 4th quartile, respectively) compared to the IL-6 KO subgroup (44.4%, 22.2%, 22.2% and 11.1% of animals in the 1st, 2nd, 3rd and 4th quartile, respectively), the difference was not statistically significant (Fischer’s exact *p* = 0.157) (Figure 2A).

Histology confirmed Grade 1 to 2, Stage pTis or T1 tubular adenocarcinoma in both groups (Appendix A). Only low-grade tumors were found in IL-6 KO mice, whereas the distribution of low and moderate grade tumors was equal in the WT group (Figure 2B).

IL-6 KO mice exhibited an increased mortality rate compared to WT animals (Figure 3). The IL-6 KO animals showed more severe inflammation after each treatment cycle registered in the decimal scoring system and therefore more animals were lost during the study before reaching the end point (Appendix A).

Platelet counts were normally distributed (Shapiro–Wilk test *p* = 0.14, *p* = 0.71, *p* = 0.98, *p* = 0.99 in the IL-6 KO control, IL-6 KO treated, WT control and WT treated animals, respectively) with equal variances across subgroups (Bartlett’s test *p* = 0.68). Mean platelet counts differed between the four subgroups (ANOVA F_3,21_ = 31.16, *p* < 0.001). Concerning the treatment groups, mean (±SD) platelet count was significantly higher (Wald test F_1,21_ = 30.56, *p* < 0.001) in the WT (754 ± 103 G/L) than in the IL-6 KO animals (340 ± 109 G/L). In both genotypes, platelet count was lower in the control than in the treatment groups and this difference was significant (Wald test F_1,21_ = 66.39, *p* < 0.001) with the wild genotype (Figure 4).

### 2.3. PCR Results

Liver thrombopoietin-1 (TPO) gene expression was higher in the WT CRC model group than in the other groups (Figure 5). The mean TPO levels were similar in the control groups and the IL-6 KO CRC model group, although the data range was greater in the latter group. On the contrary, the expression of thrombospondin gene (THBS-1) in liver was significantly elevated in the IL-6 KO group compared to the other groups. There was no difference between the control groups regarding THBS-1 expression, whereas mice in the WT treatment group showed slightly elevated expression that did not reach significance.

### 2.4. Intraluminal Fluorescent Endomicroscopy

Intraluminal endomicroscopy showed marked difference between the animals. In WT animals with tumors, after acriflavine nuclear staining, the mucosal luminal surface in vivo images showed asymmetric patterns of crypts. The opening of crypts was increased in diameter and their shape deflected from circular. Crypt walls were also seemingly more infiltrated with cells, and overall cell content was higher. Red channel revealed increased mucus production. In IL-6 KO animals, both in vivo and immediately post-harvesting crypt morphology was similar to normal with regular crypt pattern (Figure 6).

### 2.5. Imaging Studies

Positron emission tomograph (PET)/magnetic resonance imaging (MRI) images also showed clear differences between the WT and IL-6 KO animals (Figure 7). In the WT mouse (^18^F)-fluoro-deoxy-D-glucose (FDG) uptake in the descendent colon and transverse colon was very high corresponding with tumor foci (Figure 7A,B,I). In the IL-6 KO mouse large intestinal uptake remained scarce, with no signs of lymph node involvement (Figure 7C,D,G,H). Interestingly, increased peritoneal FDG activity uptake was observed and a small bowel focus was also detected (Panel E, F).

Inflammatory reactions and water signal enhancements in the WT animal were also more prominent in MRI and PET imaging, showing an enlarged retroperitoneal lymph node in MRI and its moderate–high FDG uptake (Appendix A). The simultaneous placement of mice in the multi-animal bed enabled to create an abdominal cross-sectional overview of the reconstructed PET/MRI images (Appendix A).

We compared the FDG uptake in the highly avid tumorous foci in the WT animal and the two IL-6 KO animals, by fitting a mixed effects model using a maximum-likelihood estimator, assuming fixed effect for IL-6 status and a random effect for maximal standardized uptake values (SUV_max_) by each animal. FDG uptake was significantly greater in the WT animal compared to the IL-6 KO animals (*p* < 0.001) (Figure 8).

All in all, the WT animal presented higher FDG uptake and use, concomitantly to qualitative image analysis showing more intense and larger colonic FDG uptake foci than observed in IL-6 KO animals (Table 1).

## 3. Discussion

It has been previously established that in many solid tumors, a relationship exists between thrombocytosis detected at diagnosis and tumor spread or metastasis, as well as shorter patient survival [5,6].

The principal regulator of platelet production is TPO [20,21], which is mainly controlled by receptor-mediated uptake and destruction. The regulation of plasma TPO level is closely linked to platelet count [22]. When the latter is low, less TPO is absorbed, and the elevation of TPO level leads to enhanced thrombopoiesis. KASER et al. found in mice that thrombocytosis is accompanied by the simultaneous increase in the expression of thrombopoietin mRNA by the liver, leading to elevation of plasma TPO level [23].

IL-6 is a pleiotropic cytokine that plays a role in both normal hemostasis and immune responses. Under physiologic conditions it is almost undetectable. It binds with similar affinity to both transmembrane IL-6R (mIL-6R) which is expressed only on some lymphoid (monocytes, macrophages, neutrophils, B-cells, subpopulation of T-cells) and on non-lymphoid cells (hepatocytes) and to soluble IL-6R (sIL-6R), which is an isoform of mIL-6R without cytoplasmic and transmembrane domains and can be detected in the plasma. Ninety percent of sIL-6R are generated in a process called shedding: the ectodomain of the mIL-6R is released in the extracellular space by the A Disintegrin and Metalloproteinase 17 (ADAM17). If IL-6 binds to the mIL-6R during the classic signaling pathway, it exerts anti-inflammatory effects, including induction of fever, acute phase response and the differentiation of B-cells to plasma cells. However, if IL-6 binds to the sIL-6R, their complex can be bound to the transmembrane signal transducer protein gp130, which is ubiquitously expressed on all cell types, including tumor cells. This pathway is called trans-signaling, and it has proinflammatory effects, such as maintaining of Th17 phenotype in inflamed tissues, inhibition of lamina propria T-cell apoptosis and the malignant proliferation of epithelial cells.

Elevated IL-6 levels have been observed in a variety of tumors, including gastrointestinal cancer [24]. Similar to thrombocytosis, IL-6 was also found to correlate with tumor stage, size, metastasis and patient survival in CRC. Both tumor cells themselves and immune cells infiltrating tumor tissue produce IL-6 [25,26].

Stone et al. studied ovarian cancer patients with thrombocytosis [15] and determined the plasma levels of several thrombopoietic factors (TPO, IL-1α, IL-3, IL-4, IL-6, IL-11, G-CSF, M-CSF, SCF and Flt3-ligand). The plasma levels of TPO and of IL-6 correlated significantly with platelet count: patients with thrombocytosis had significantly higher IL-6 and TPO plasma levels, moreover, at given cutoff values, plasma IL-6 level and thrombocytosis have both been found to be independent prognostic factors. Based on these results, a tentative, paracrine-mediated, paraneoplastic pathway has been postulated in which IL-6 expressed and secreted by the ovarian tumor would increase TPO production by the liver. This in turn would stimulate the bone marrow and eventually elevate platelet count with an end-result of tumor-induced thrombocytosis.

In our AOM-DSS-induced CAC model we found significantly higher platelet counts in the WT CRC group than in the WT control or the IL-6 KO groups. Parallelly, increased liver TPO-1 expression was detected in the WT CRC group compared to the other groups.

Data obtained in our model are consistent with other reports presenting IL-6 KO mouse constructs and various roles of IL-6 in behavior, systemic inflammation, and most importantly, in the course of inflammation-related local tumor induction [27,28,29]. As in other reports, our results also show more homogenous and lower grade tumors in IL-6 KO animals than in WT animals. The systematically positive effect of IL-6 on tumor survival must be accounted for: IL-6 binds to soluble IL-6R (sIL-6R), which interacts with gp130 on tumor cells. This activates Janus kinases resulting in the phosphorylation of signal transducer and activator transcription 3 (STAT3). STAT3 induces target gene transcription in the nucleus leading to proliferation, cell growth and the inhibition of apoptosis [30]. What is unfavorable during tumor development is favorable in acute inflammation. Expression analysis of intestinal epithelial cells showed that ADAM17 was highly upregulated on tumor tissue while the mIL-6R was strongly downregulated [27,31]. In a murine model of colitis-associated premalignant cancer, elevated expression of ADAM17, IL-6, gp130 and sIL-6R in lamina propria macrophages were reported [31]. It suggests that these cells shed the IL-6R from the cell surface, and at the same time they were the source of IL-6 (Figure 9).

Surprisingly, the IL-6 KO animals had an increased mortality in our model due to intestinal inflammation being more severe than in WT animals. After the third cycle of DSS administration 50% of IL-6 KO animals were lost (*n* = 9), whereas only one WT mouse had lethal outcome due to the severity of the inflammation.

Several reports are in in line with our results: if IL-6/gp130/STAT3 pathway is impaired and mice are exposed to AOM-DSS, more severe colitis with more pronounced epithelial damage and ulceration, more prominent infiltration with inflammatory cells can be observed, while tumor load is decreased due to a reduction in size and frequency [26,27,28]. Irrespectively from the type of injury (infectious or non- infectious), epithelial proliferation and repair are impaired, and the host is more susceptible to mucosal damage if IL-6 is inhibited [32,33].

At the endpoint we performed imaging studies in animals from both treatment groups. By this time, the acute inflammation subsided significantly. However, IL-6 can maintain chronic inflammation by affecting the differentiation and survival of pathogenic T helper cells. Therefore, at this time point the WT animal showed signs of chronic inflammation and increased tumor burden simultaneously. The limitation of the findings during in vivo combined microscopy and PET/MRI imaging is, that only three animals were examined. The differences between the groups are marked, but it cannot be ruled out, that certain characteristics may be found only in the individual animal. The WT animal showed characteristics of mucosal inflammation during endomicroscopy, whereas PET/MRI presented increased FDG uptake in the large intestines, peritoneum and lymph node. Furthermore, the overall FDG uptake was significantly higher in the WT animal. The increased focal small bowel uptake and slightly elevated peritoneal uptake in the IL-6 KO animal may refer to a non-healing inflammatory focus and the consecutive irritation of the peritoneum.

In conclusion, we demonstrated significantly elevated platelet counts and TPO expression in WT animals compared to IL-6 KO animals, which supports the previously proposed paracrine paraneoplastic pathway. We have also shown the importance of IL-6 not only in tumor development, but also in the anti-inflammatory and reparative processes during colitis.

## 4. Materials and Methods

### 4.1. IL-6 Gene Knockout

A new IL-6 knockout mouse strain was established using the CRISPR/CAS9 method. Further details on the applied molecular biologic techniques to produce this mouse strain are available in the Appendix A.

### 4.2. Animals Used in the Experiments

Adult (8–10-week-old) male FVB/Ant and IL-6 KO FVB/Ant mice were then obtained from the local colony bred at the Medical Gene Technology Unit (specific pathogen-free (SPF) level) at the Institute of Experimental Medicine in Budapest, Hungary. Animals were housed at the Minimal Disease (MD) level, 3–5/cage, under controlled environmental conditions: temperature, 21 °C ± 1 °C; humidity, 65%; light–dark cycle, 12-h light/12-h dark cycle, with the lights switched on at 07:00 a.m. Mice had free access to rodent food and drinking water. A DSS solution admixed to water was provided according to the experimental protocol. All procedures were conducted in accordance with the guidelines by the European Communities Council (86/609/EEC/2 and 2010/63 Directives of European Community) and the protocol of the experiment was approved by the Institutional Animal Care and Use Committee of the Institute of Experimental Medicine, Budapest, Hungary (8 May 2013) with the permit number: PEI/001/29–4/2013.

### 4.3. Experimental Protocol

We used the AOM [34]/3% DSS model to induce inflammation-based colorectal cancer. AOM is a chemical agent, which initiates neoplastic transformation by DNA methylation and thereby facilitates base mispairing. DSS is a sulfated polysaccharide of variable molecular weight that induces inflammation, resembling ulcerative colitis in humans [35,36]. At baseline, the mice received a single AOM dose (7.4 mg/kg BW) by the intraperitoneal route. This was followed by 3 cycles of treatment with DSS-containing water for one week and then, administering normal drinking water for two weeks (Figure 10). During the 9-week treatment period the weight, water consumption, fecal consistency and condition of the perianal region of the animals were recorded regularly. The severity of the inflammation was estimated based on these measurements and it was presented in a decimal scoring system. At the end of the experiment, the animals were decapitated, their blood was collected in pre-cooled EDTA tubes, and the plasma was stored at −20 °C until analysis. Liver and colon tissue were obtained for histology, and some of the specimens were frozen on dry ice for RT–PCR. In each animal the complete colon was examined both macroscopically and under microscope with hematoxylin–eosin staining by a single pathologist to control inflammation, tumor formation and stage.

### 4.4. Quantitative Real-Time PCR

Frozen blood and liver tissue samples were homogenized in TRI Reagent Solution (Ambion, USA) and total RNA was isolated with QIAGEN RNeasy Mini Kit (Qiagen, Valencia, CA, USA) according to the manufacturer’s instructions. To eliminate genomic DNA contamination, DNase I treatment was used and 100 μL RNAse-free DNase I (1 unit DNase) (Thermo Scientific, Waltham, MA, USA) solution was added. Sample quality control and the quantitative analysis were carried out by NanoDrop 2000 (Thermo Scientific). Amplification was not detected in the RT-minus controls. The cDNA synthesis was performed with the high capacity cDNA reverse transcription kit (Applied Biosystems, Waltham, MA, USA). Primers for the comparative Ct experiments were designed by Primer Express 3.0 Program and Primer Blast software. The primers (Microsynth, Balgach, Switzerland) were used in the real-time PCR reaction with Fast EvaGreen^®^ qPCR Master Mix (Biotium, Fremont, CA, USA) on ABI StepOnePlus instrument. Gene expression was analyzed with the ABI Step One 2.3 program (https://www.thermofisher.com/hu/en/home/technical-resources/software-downloads/StepOne-and-StepOnePlus-Real-Time-PCR-System.html). The amplicon was tested by Melt Curve Analysis on ABI StepOne Plus Instrument (https://www.thermofisher.com/hu/en/home/technical-resources/software-downloads/StepOne-and-StepOnePlus-Real-Time-PCR-System.html). The experiments were normalized to *rplp1* expressions.

The following (forward and reverse) primer pairs were used:rplp1 for TAAGGCCGCGTTGAGGTGrplp1 rev GATCTTATCCTCCGTGACCGTthbs1 for CAT GCC ATG GCC AAC AAA CAthbs1 rev TTG CAC TCA CAG CGG TAC ATthpo1 for CTT CTC CAC CCG GAC AGA GTthpo1 rev CTG GCC AGG GTG TCT AAC TG

### 4.5. In Vivo Imaging Using PET/MRI and Intraluminal Fluorescent Confocal Endomicroscopy

Colonic mucosal surface was imaged in vivo, using intraluminal fluorescence confocal endomicroscopy. Eventual differences in colonic mucosal crypts in WT and IL-6 KO mice were imaged with a nuclear counterstaining and mucus autofluorescence. One WT mouse and two KO mice were anesthetized using 3% *v/v* isoflurane gas (Isofluran-KP^®^, Medicus Partner, Ltd., Budaörs, Hungary) and maintained on a heated bed. Then, after 1 mL of saline enema application, a 6% *m/v* acriflavine solution in water for injection (Sigma-Aldrich Ltd., Budapest, Hungary) in 0.5 mL volume was rectally applied to each mouse. Acriflavine, a green fluorescent DNA-binding molecule is a general cell-staining intravital dye to obtain fluorescent contrast of luminal mucosal cells. Thereafter, the S1500 fiberoptic microscope probe tip (resolution 3.3 microns, field-of-view diameter, 600 microns) of the Cellvizio Lab Dual Band imaging system (Mauna Kea Technologies, Inc., Paris, France) was intrarectally applied into the descending colon of the mice and dual band (480 nm green and 660 nm red) live microscopic images of the mucosal walls were collected. After terminal euthanasia of the same animals with Euthasol ad us. vet solution (Medicus Partner, Budaörs, Hungary) colons were opened, and mucosal surfaces were investigated ex vivo also within the same animals. Cells in crypts fluoresce in green, with an especially prominently stained nucleus. Mucus in the crypts can be present as red fluorescent dots (from autofluorescence).

To further assess whole-body tumor and inflammatory burden, one WT and two IL-6 KO mice were used in a series of imaging experiments.

Mice were first scanned with a dedicated nanoScan^®^ PET/MRI system (Mediso, Ltd., Budapest, Hungary) for small animals with a 3 Tesla magnetic field, using FDG as probe for increased glucose transport and utilization rate and glucose transporter protein expression in cells of tumorous or inflamed tissue foci. The MRI imaging sequence was chosen with regards to increased water concentration of edematous inflamed tissues. Water signals in lymph nodes or intestinal walls are visualized as areas with lower brightness in gray (among brighter fat signals).

Sixty minutes prior to imaging mice were intraperitoneally injected with 3–4 MBq of FDG solution (FDG-KEDOI^®^, Pet-Medicopus, Ltd., Kaposvár, Hungary). Then, mice were anesthetized with 5% isoflurane as an induction and 2% as maintenance inhalation dose in medical oxygen. Mouse PET/MRI measurements were performed in a MultiCell™ heating, positioning and monitoring multi-animal bed for simultaneously holding 3 mice in the same imaging session (Mediso, Ltd., Budapest, Hungary). Ninety minutes after FDG injection, a 20-minute static PET data acquisition was obtained, immediately followed by a 2D spin echo MRI sequence with 0.3 mm slice thickness to obtain anatomic background. Quantitative radioactivity PET data were reconstructed using a Monte-Carlo-based iterative algorithm (Tera-Tomo™, Mediso, Ltd., Budapest, Hungary) using the MRI as anatomic and attenuation priors, with 0.3-mm voxel size for a whole-body PET/MRI image with a resolution of 1 mm.

### 4.6. Image Analysis

For the Cellvizio images, qualitative analysis of crypt forms (green channel) and red mucus dots were performed by three of the authors (N.H., D.S.V., D.M.) in a blinded fashion.

The PET/MRI images were qualitatively assessed from a radiologic point of view and enlarged lymph nodes or bowel luminal tumors were identified using both the MRI and PET images. PET images were then quantitatively analyzed with the vivoQuantTM software (inviCRO, Ltd., Boston, USA) by defining the whole intestinal volume in a three-dimensional volume of interest and calculating the standardized FDG uptake value and the radioactivity concentration percentage to whole body radioactivity for each animal in the intestinal volume of interest.

### 4.7. Statistical Analysis

We compared tumor growth in the IL-6 KO and WT subgroups AOM/DSS-treated CRC model mice in terms of the number of tumor foci and TTV. TTV was calculated by adding the volume of all tumor foci within each animal. We tested the normality of distributions using the Shapiro–Wilk test. In case of normal distribution and equal variances, we compared subgroups via the Student’s *t*-test. If we detected non-normal distribution in either subgroups, we performed nonparametric tests, such as the Kolmogorov–Smirnov test, the Mann–Whitney U test and the equality of medians test. When applicable, *p*-values were determined using exact methods. We also grouped animals into TTV quartiles and compared their distribution between IL-6 KO and WT subgroups using cross-tabulation and the Fischer’s exact test. Furthermore, platelet count was compared via ANOVA. Pairs of subgroups were compared using Wald–tests on OLS regression coefficients.

We quantified the radiographic results of a WT and two IL-6 KO animals by comparing the FDG uptake in the three most avid tumorous foci displaying highest SUVmax per animal. We fitted a mixed effects model using a maximum-likelihood estimator, assuming fixed effect for IL-6 status and a random effect for SUVmax by each animal. Statistical analysis was conducted using Stata version 14 (StataCorp. 2015. Stata Statistical Software: Release 14. College Station, TX: StataCorp LP. https://www.stata.com/support/faqs/resources/citing-software-documentation-faqs/)

## Figures and Tables

**Figure 1 ijms-21-06218-f001:**
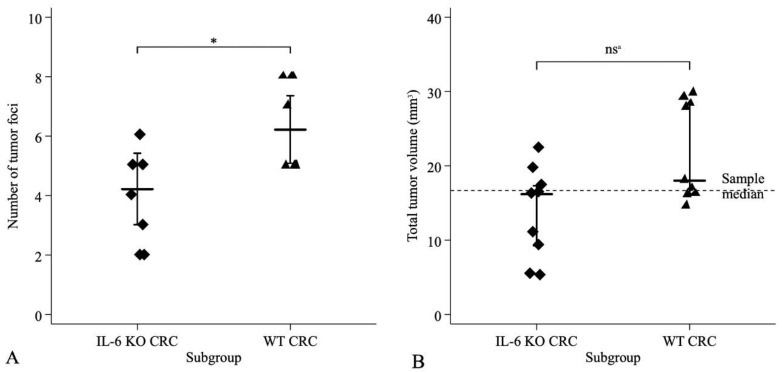
Distribution of (**A**) tumor foci and (**B**) total tumor volume (TTV) in IL-6 KO CRC model and WT CRC model groups. Total number in both groups was *n* = 9. A—number of tumor foci with two-sided Student–test; B—TTV with nonparametric tests. * Two-Sided Student’s *t*-test: *t*_16_ = −2.79, *p* = 0.013. ns^a^ (^a^ not significant) Kolmogorov-Smirnov test exact *p* = 0.352, Mann-Whitney test *p* = 0.058, non-parametric equality of medians test exact *p* = 0.347.

**Figure 2 ijms-21-06218-f002:**
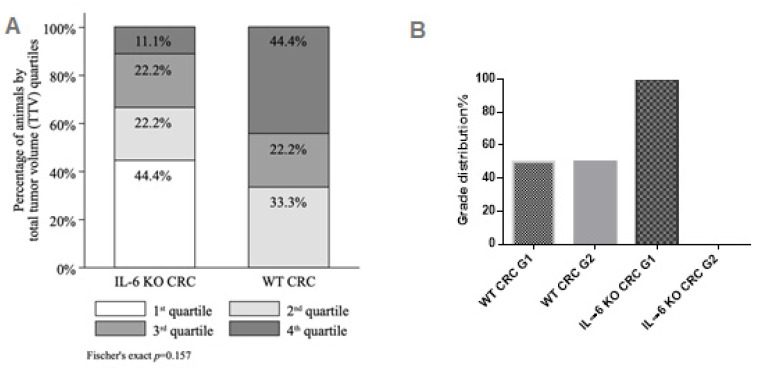
(**A**) Total tumor volume (TTV) quartiles by IL-6 status (IL-6 KO CRC *n* = 9, WT CRC *n* = 9) with Fisher’s exact test; (**B**) tumor grade distribution.

**Figure 3 ijms-21-06218-f003:**
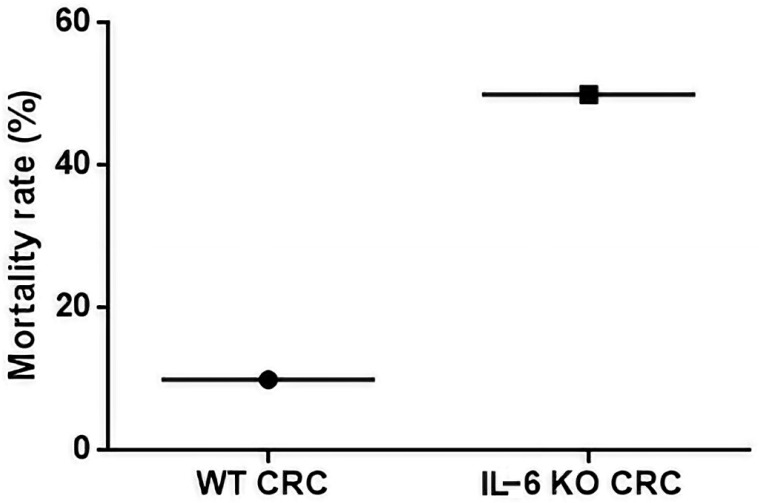
Mortality rate in WT CRC and IL-6 KO CRC groups (WT, IL-6 KO *n* = 9, respectively). Mortality rate was calculated as the number of dead animals at Week 12 divided by the total number of animals at Day 1.

**Figure 4 ijms-21-06218-f004:**
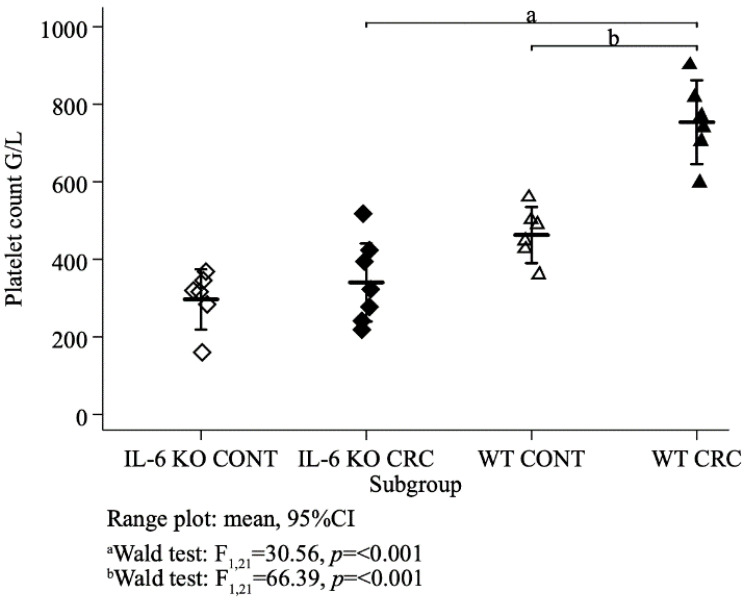
Mean platelet counts of all animal groups (G/L). (IL-6 KO CONT *n* = 6, IL-6 KO CRC *n* = 7, WT CONT *n* = 6, WT CRC *n* = 6,) ordinary least squares (OLS) regression, Wald test.

**Figure 5 ijms-21-06218-f005:**
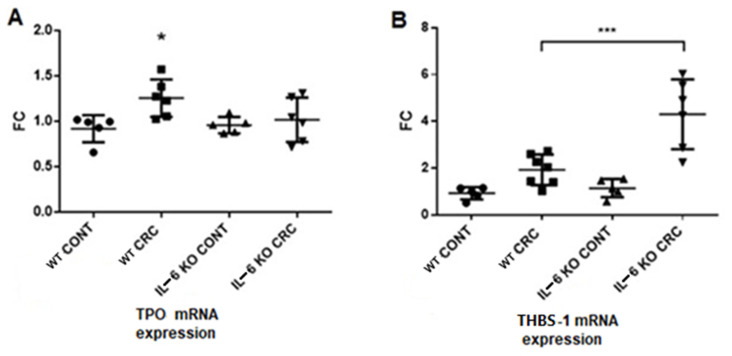
(**A**) TPO and (**B**) THBS-1 mRNA expressions in the liver. mRNA expressions were measured by quantitative real-time PCR. (WT CONT *n* = 5, WT CRC *n* = 6, IL-6 KO CONT *n* = 5, IL-6 KO CRC *n* = 6) Data represent the mean (SD). (**A**): * *p* < 0.05 (B): *** *p* < 0.001 (FC = fold change).

**Figure 6 ijms-21-06218-f006:**
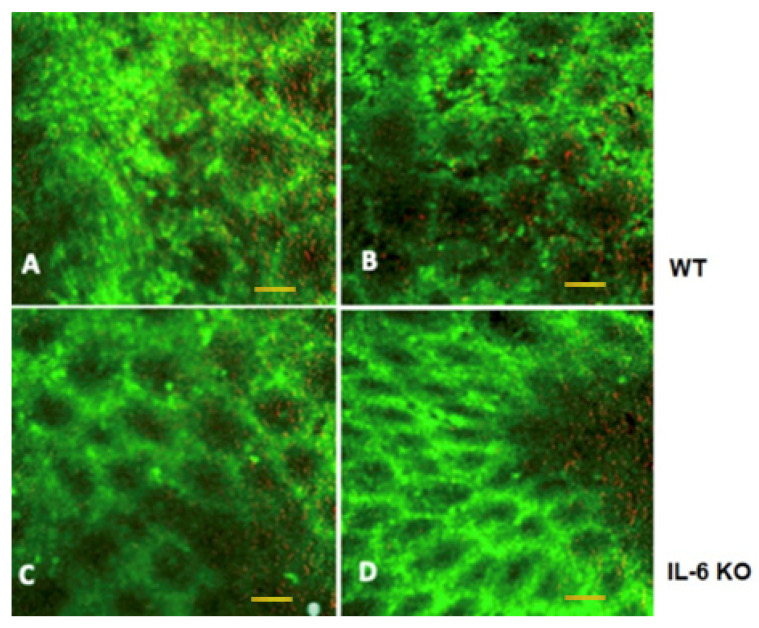
Intraluminal fluorescent endomicroscopic images of WT and IL-6 KO mouse colon. (**A**) In vivo intraluminal image; (**B**) ex vivo mucosal surface image in a WT mouse with colonic tumor; (**C**) in vivo intraluminal image; (**D**) ex vivo mucosal surface image in an IL-6 knockout mouse with inflammation-related colonic neoplasia. 488 nm (green) and 660 nm (red) dual band fluorescent Cellvizio endomicroscopic images. Scale bar represent 50 microns for all images.

**Figure 7 ijms-21-06218-f007:**
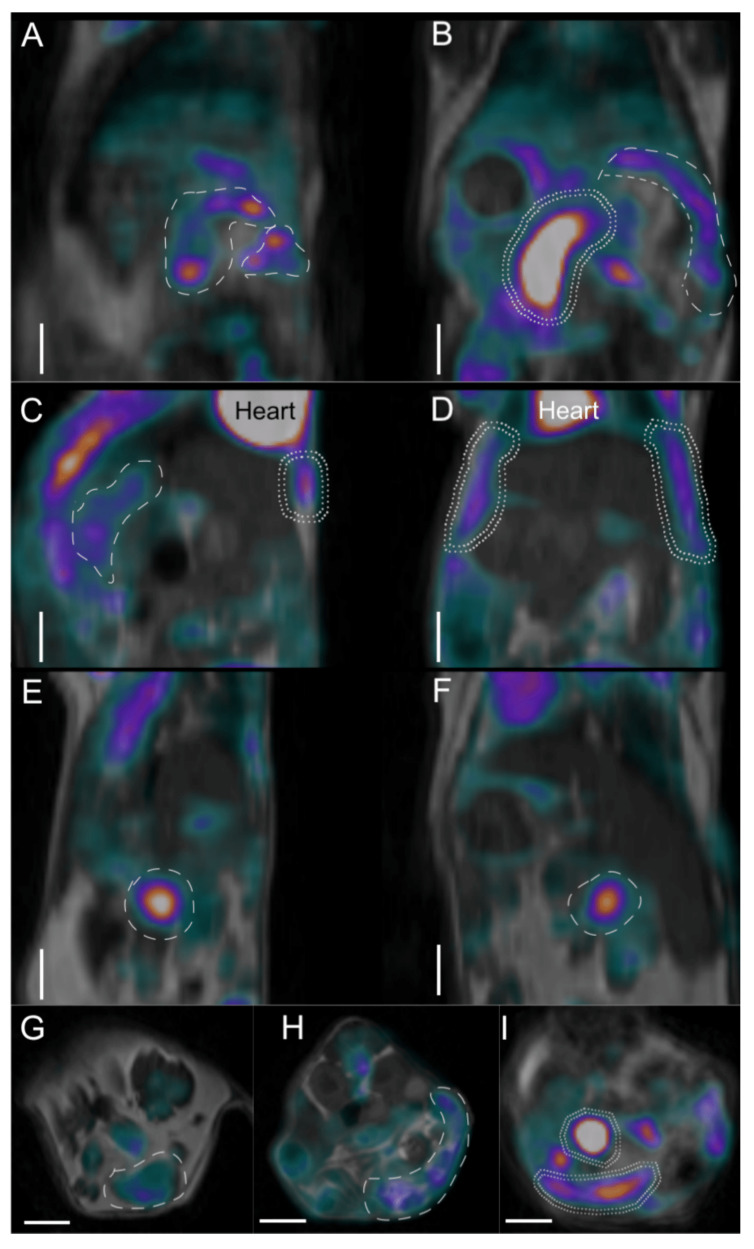
Positron emission tomograph (PET)/magnetic resonance imaging (MRI) imaging of WT (**A**,**B**,**I**) and IL-6 KO (**C**–**H**) mouse abdominal regions in three section planes. (**A**) Sagittal planes of WT mouse; (**B**) horizontal section of WT mouse; (**C**,**E**) sagittal planes of IL-6 KO mice; (**D**,**F**) horizontal planes of IL-6 KO mice; (**G**,**H**) cross-sections of IL-6 KO mice; (**I**) cross-section of WT mouse abdomen. High [18F]-fluoro-deoxy-d-glucose (FDG) activity uptake regions (over 0.2 standardized uptake values (SUV values) are circumscribed with dashed lines. Double-dashed line shows very high (over 0.4 SUV value) FDG uptake regions. Color scales are grayscale for the background anatomic MRI images, while the PET component shows low uptake in black (no uptake) to blue (low uptake), moderate in purple to red and higher FDG uptake in red through yellow to white (highest uptake). Heart is annotated in panels C and D for reference. White scale bars in each panel equal 500 microns.

**Figure 8 ijms-21-06218-f008:**
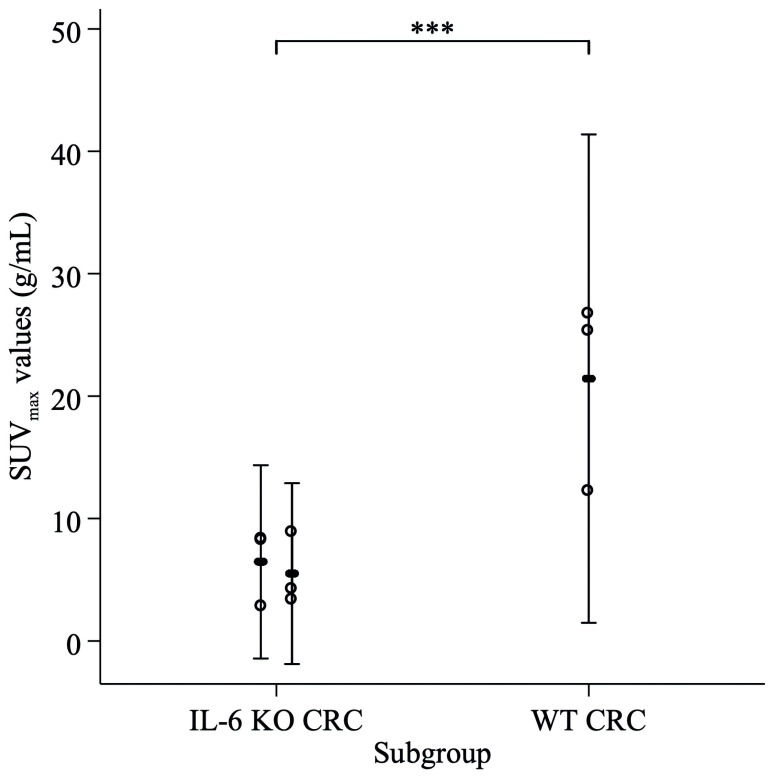
FDG uptake comparison in the highly avid tumor foci in the WT animal and the two IL-6 KO animals, mixed effects model. Hollow markers: SUV_max_ values of three most avid focal regions. Range plot: SUV_max_ mean, 95%CI by animal. *** Mixed effects model, fixed effect of IL-6 status (IL-6 KO vs. WT): *p* < 0.001.

**Figure 9 ijms-21-06218-f009:**
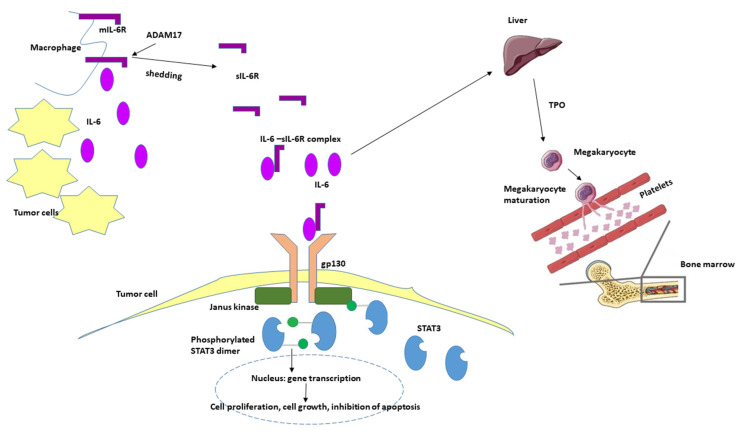
Hypothetical role of IL-6 in oncogenesis and tumor-associated thrombocytosis. IL-6 is produced in the tumor microenvironment both by mononuclear and tumor cells. Elevated levels of IL-6 induce thrombopoietin production in the liver, which in turn activates the maturation of megakaryocytes and the production of platelets resulting elevated platelet count. Simultaneously, IL-6 binds to the soluble form of IL-6 receptor (sIL-6R) and activates the gp130/JAK/STAT3 pathway leading to the translocation of phosphorylated STAT3 homodimer to the nucleus in the tumor cells and to the transcription of different genes inducing proliferation, cell growth and the inhibition of apoptosis. mIL-6R—membrane-bound IL-6 receptor, TPO—thrombopoietin, ADAM17—a disintegrin and metalloproteinase 17; STAT3—signal transducer and activator transcription 3.

**Figure 10 ijms-21-06218-f010:**
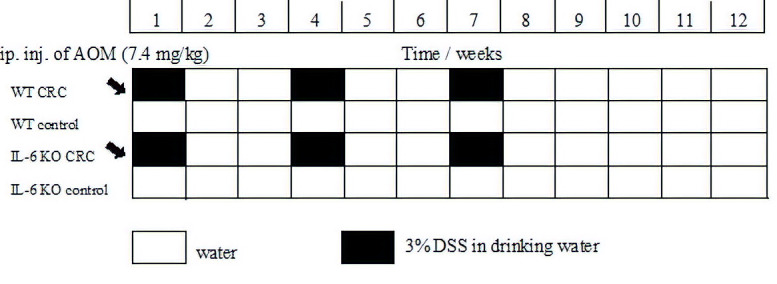
Timeline description of colorectal cancer induction using the AOM/DSS inflammation-related model. AOM—azoxymethane; DSS—dextran sodium sulfate; water—drinking water. Black arrow: administration of AOM intraperitoneally.

**Table 1 ijms-21-06218-t001:** PET FDG-uptake differences. PET FDG-uptake differences in the intestinal volume of interest containing tumorous foci in imaged mice.

Animal	Averaged Intestinal Standardized Uptake Value of Intestines (g/mL)	Radioactivity Content in Intestines vs. Whole-Body Radioactivity
Mouse 1 WT CRC	0.43	20.1%
Mouse 2 IL-6 KO CRC	0.15	5.3%
Mouse 3 IL-6 KO CRC	0.27	6.8%

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
