# Peer review of "Thrombocytosis and Effects of IL-6 Knock-Out in a Colitis-Associated Cancer Model"

_ijms, 2020, doi:10.3390/ijms21176218_

Round 1

Reviewer 1 Report

In their manuscript, Josa and co-authors use a CRISPR/cas9 IL-6 knockout (KO) to evaluate the role of IL-6 in tumor development and thrombocytosis in a model of inflammation-induced colorectal cancer.

General consideration

This is an interesting work about a relevant issue, and the comprehensive methodology applied to answer the scientific questions seems appropriate, although the statistical relevance of some of the observations is weak.

Some minor issues affect the quality of the paper, that can be improved mostly by making the message more clear-cut. In particular, the Discussion section start with a very long literature review that seems to be more suitable for the introduction, also because it lacks a direct comparison with the experimental results of the study. In general, the Discussion does not offer a clear picture of the presented results and the message is not clear. For example, the relationship between increased systemic inflammation and reduced tumor burden in IL6ko mice is not extensively discussed and does not help understanding pros and cons of the presented model.

Specific issues:

Figure legend should contain a clear indication of how many animals were used in the indicated experiment. The reviewer is aware that working with transgenic animals implies some technical limitations, but readers need to know the statistical significance of the results.

Imaging studies need to be enriched by adding at least one or two animals per group, otherwise it’s hard to say whether the differences observed between WT and IL6KO animals just reflect normal fluctuation among different subjects.

Something happened in the Methods “Image Analysis” section, where it seems that part of the Abstract (not shown above) shows up.

Overall, the clarity of the message need to be improved

Author Response

Point 1: English language and style are fine/minor spell check required

Response: The manuscript was reviewed by a native speaker.

Point 2: Discussion section start with a very long literature review that seems to be more suitable for the introduction, also because it lacks a direct comparison with the experimental results of the study. In general, the Discussion does not offer a clear picture of the presented results and the message is not clear. For example, the relationship between increased systemic inflammation and reduced tumor burden in IL6ko mice is not extensively discussed and does not help understanding pros and cons of the presented model.

Response: The Discussion section has been completely revised. IL-6 KO mice showed reduced tumor development, however, at the same time they reacted to DSS with a more severe colitis, often leading to death.

Point 3: Figure legend should contain a clear indication of how many animals were used in the indicated experiment.

Response: Corrected.

Point 4: Imaging studies need to be enriched by adding at least one or two animals per group, otherwise it’s hard to say whether the differences observed between WT and IL6KO animals just reflect normal fluctuation among different subjects.

Response: In the short time available for review, it was not possible to restart the IL-6 KO mouse strain from the frozen embryos (also coupled to partly working animal house services due to the COVID-19 situation).The imaging studies have been showing and reinforcing the identical findings of the macroscopic findings and microscopic analysis made with the use of a substantial number of animals, therefore they are organically embedded to the study even with a few animals (which hindrance now has nevertheless been further mentioned in our discussion, too). The magnitude and qualities of the differences both in PET/MRI and in the intraluminal dual wavelength confocal laser endomicroscopy, interpreted together with the other in vivo results are difficult to appropriate to a fluctuation among different subject. This is shown in Table 1 second row, as the nearly 5-fold intestinal total radioactivity proportional uptake difference is a sign of underlying inflammatory difference and could be very hardly interpreted as intersubject variance. To further substantiate this claim, we have added another measurement of maximal tumorous Standardized Uptake Values (SUVmax) in the three highest visually identified FDG-avid focal volumes of interest of the intestinal region as Table 1 of Supplementary Information. This table, approximately 4-fold increase of WT animals within, also presents a convincingly distinct difference among the IL-6 KO and the WT strains in elevated tumor glucose metabolism, which is characterised with SUVmax in translational PET imaging. However, this weakness of the imaging findings is discussed in the revised paper.

Point 5: Something happened in the Methods “Image Analysis” section, where it seems that part of the Abstract (not shown above) shows up.

Response: This was maybe attributable some problems during uploading. It has been deleted from the image analysis.

Reviewer 2 Report

The experimental data are too preliminary. It is essential point in this study, whether IL6-KO suppressed carcinogenesis or not. However, figures 2 and 3 show ambiguous results. Furthermore, figure 1 is hard to understand.

Author Response

Point 1: Moderate English changes required

Response: The manuscript was reviewed by a native speaker.

Point 2: The experimental data are too preliminary.

Response: We think our results are forward-looking.

Point 3: It is essential point in this study, whether IL6-KO suppressed carcinogenesis or not.

Response: The discussion has been completely revised. Our study shows that IL-6 plays an important role in carcinogenesis. IL-6 KO mice exhibited significantly reduced tumor development (fewer and smaller tumors), however, they developed more severe inflammation after DSS administration. They showed much worse mortality rate than WT animals, however, this was attributable to the inflammation. IL-6 activates gene transcription in cells resulting proliferation, cell growth and inhibition of apoptosis. These effects are favorable during cell repair in inflammation, but unfavorable in tumor development.

Point 4: Figure 1 is hard to understand.

Response: Figure 1 was edited (during uploading the figure underwent changes and this has been corrected).

Round 2

Reviewer 1 Report

The reviewers completely understand the difficulties behind performing the new experiments requested. The current form of the manuscript is actually more clear and straightforward.

Author Response

Please see the

Dear Editors and Reviewers,

We are grateful for the comments of the reviewers which have helped to improve the quality of our manuscript.

We have made all the requested changes, the complete manuscript has been reviewed by a native speaker again. Please, read our point by point response below.

The revised texts were made with Track changes.

We have improved the quality of the figures in the manuscript. We have uniformized the figures regarding their legends. We have changed the order of the manuscript as required by the journal.

The reference sample that can be found on the journal’s website differs from sample that can be downloaded from Endnote, e.g. the doi is present in the Endnote sample. Therefore, we have attached the Endnote file.

We hope that the proposed work is now acceptable for publication.

All the authors read and approved the changes.

Sincerely yours,

Valeria Josa

Response to Reviewer 1 Comments

Point 1: English language and style are fine/minor spell check required

Response: The manuscript has been reviewed by a native speaker.

Point 2: The reviewers completely understand the difficulties behind performing the new experiments requested. The current form of the manuscript is actually more clear and straightforward.

Response: We thank the Reviewer for his opinion.

Reviewer 2 Report

Point 2: The experimental data are too preliminary.

Response: We think our results are forward-looking.

Re:

Usually, preliminary animal experiments are done to search adequate condition, with using small number of mice. And then, the reproducibility of the results should be confirmed under the optimized condition, with using enough number of mice for statistic analysis.

As the authors say, IL-6 KO mice seemed to reduce tumor development (fewer and smaller tumors), however, they developed more severe inflammation after DSS administration. In this study, the number of cycles of DSS-treatment should be decreased to avoid severe inflammation by DSS, and the experimental period may be extended if tumorigenesis is not enough. It is very interesting point the relationship of "tumor-related inflammation" and IL6 and thrombocytosis. The authors should statistically show the reduction of "tumorigenesis" or "carcinogenesis" by IL6-KO.

Point 4: Figure 1 is hard to understand.

Response: Figure 1 was edited (during uploading the figure underwent changes and this has been corrected).

Re:

The period of administration should be clearly shown. Please learn other figures " Int. J. Mol. Sci. 2017, 18, 1750; doi:10.3390/ijms18081750 (Fig.9) " and " Int. J. Mol. Sci. 2016, 17, 1343; doi:10.3390/ijms17091343 (Fig.1) ", for example.

Please notice follows;

  1. The suitable term "tumor" or "cancer", should be determined by authors' results.
  2. Tumor-related (or cancer-related) inflammation and IL6 should be discussed.
  3. The reproducibility of the results is important. For example, figures 2 and 3 show ambiguous meaning. The authors should statistically analyze "tumor number, size, or cancer size of all animal groups", in similar manner of figure 4 of "platelet number of all animal groups". Imaging studies should be statistically analyzed, and then representative images should be clearly shown with suitable resolution in figures.
  4. It is better to show additional figure of authors' hypothesis based on the results and references.

Author Response

Dear Editors and Reviewers,

We are grateful for the comments of the reviewers which have helped to improve the quality of our manuscript.

We have made all the requested changes, the complete manuscript has been reviewed by a native speaker again. Please, read our point by point response below.

The revised texts were made with Track changes.

We have improved the quality of the figures in the manuscript. We have uniformized the figures regarding their legends. We have changed the order of the manuscript as required by the journal.

The reference sample that can be found on the journal’s website differs from sample that can be downloaded from Endnote, e.g. the doi is present in the Endnote sample. Therefore, we have attached the Endnote file.

We hope that the proposed work is now acceptable for publication.

All the authors read and approved the changes.

Sincerely yours,

Valeria Josa

Response to Reviewer 2 Comments

Point 1: Extensive editing of English language and style required.

Response: The manuscript has been reviewed by a native speaker.

Point 2: Usually, preliminary animal experiments are done to search adequate condition, with using small number of mice. And then, the reproducibility of the results should be confirmed under the optimized condition, with using enough number of mice for statistical analysis. As the authors say, IL-6 KO mice seemed to reduce tumor development (fewer and smaller tumors), however, they developed more severe inflammation after DSS administration. In this study, the number of cycles of DSS-treatment should be decreased to avoid severe inflammation by DSS, and the experimental period may be extended if tumorigenesis is not enough. It is very interesting point the relationship of "tumor-related inflammation" and IL6 and thrombocytosis. The authors should statistically show the reduction of "tumorigenesis" or "carcinogenesis" by IL6-KO.

Response: The AOM/DSS model is a well-established mouse model for the study of colitis-associated cancer. While most mouse models require long-term treatment (several months), in the AOM/DSS model tumor development can be observed already within 10 weeks. It is reported to be a valuable platform to evaluate the pathogenesis of CAC. Several authors describe it step-by-step. The exact doses can be varied, but most authors agree to give 10-12 mg/kg AOM and 2.5-3% (w/v) DSS (Parang 2017, Ameet 2012, Neufert 2007, Okayasu 1996, Becker 2005). DSS administration used by most authors is as follows: 1 week DSS is followed by 2 weeks water to leave enough time for the mice to recover. Altogether 3 cycles of DSS are used. While the point of milder though longer inflammation and IL6 contribution is certainly an interesting next step nevertheless we think it would diverge from our initial scientific question that is based on the ‘standard’ AOM/DSS severity model and makes our results already comparable into the context of molecular medicine studies dealing with inflammation-induced CRC, adding a new perspective by investigating platelets. We think the results were enhanced now with the addition of new statistical considerations upon the Reviewer’s suggestions. We hope that our results can be published, and other research groups will be informed about our results. It would be great if other researchers also confirmed our findings, as our study design and data are reproducible.

Statistical analysis was performed regarding the tumor foci and total tumor burden (Line 194-220). The reduction is statistically not significant, but the figures show the clear difference between the CRC subgroups. We also added a new figure with total tumor volume quartiles to highlight the difference (Line 98-103, Figure 2A).

Point 4: Figure 1 is hard to understand.

Response: Figure 1 was edited (during uploading the figure underwent changes and this has been corrected).

Re: The period of administration should be clearly shown. Please learn other figures " Int. J. Mol. Sci. 2017, 18, 1750; doi:10.3390/ijms18081750 (Fig.9) " and " Int. J. Mol. Sci. 2016, 17, 1343; doi:10.3390/ijms17091343 (Fig.1) ", for example.

Response: We thank the Reviewer for his suggestion. The Figure was edited according to one of the samples, and certainly it can be better interpreted now.

Please notice follows;

  1. The suitable term "tumor" or "cancer", should be determined by authors' results.

Response: In the present study mice develop adenocarcinoma (Line 104) which is cancer. Therefore, we used tumor and cancer in an interchangeable manner.

  1. Tumor-related (or cancer-related) inflammation and IL6 should be discussed.

Response: The role of IL-6 and its different effects on membrane-bound and soluble IL-6 receptors are more extensively discussed in the Discussion section (Line 341-353, and 382-390)

  1. The reproducibility of the results is important. For example, figures 2 and 3 show ambiguous meaning. The authors should statistically analyze "tumor number, size, or cancer size of all animal groups", in similar manner of figure 4 of "platelet number of all animal groups".

Response: We thank the Reviewer’s suggestions. We performed statistical analysis of tumor foci and total tumor volume by the subgroups (Line 194-220). We described the statistical analysis of the platelet counts of the subgroups in a more detailed manner, the legends were uniformized (Line 240-46).

  1. Imaging studies should be statistically analyzed, and then representative images should be clearly shown with suitable resolution in figures.

Response: Statistical analyses were conducted on the quantitated results of PET (MRI). The SUV values were compared among the animals. The WT animal showed significantly greater 18F-FDG uptake compared to the IL-6 KO animals. As per our knowledge the physical resolution of the applied specific rodent imager nanoScan PET/MR 3T system is one of the highest currently worldwide with PET; 700 microns. However it follows from molecular imaging in vivo system design principles and physical limits that these small animal systems are not microscopic in imaging. We enhanced the graphical resolution of our images to 600 dpi, maybe during uploading the required resolution was lost.

  1. It is better to show additional figure of authors' hypothesis based on the results and references.

Response: We created a figure about our hypothesis (Line 391).

Round 3

Reviewer 2 Report

The new figure 1 is important analysis. It will be useful for other researcher.

The new figure 9 is not based on the results. IL-6 is induced by DSS but not by oncogenesis in this study. The figure title should include "hypothesis".